# Control Strategies to Cope with Late Wilt of Maize

**DOI:** 10.3390/pathogens11010013

**Published:** 2021-12-23

**Authors:** Ofir Degani

**Affiliations:** 1Plant Sciences Department, MIGAL—Galilee Research Institute, Tarshish 2, Kiryat Shmona 11016, Israel; d-ofir@migal.org.il; 2Faculty of Sciences, Tel-Hai College, Upper Galilee, Kiryat Shemoneh 12210, Israel

**Keywords:** *Cephalosporium maydis*, chemical control, crop protection, fungus, *Harpophora maydis*, *Magnaporthiopsis maydis*, real-time PCR, *Trichoderma*

## Abstract

Control of maize late wilt disease (LWD) has been at the forefront of research efforts since the discovery of the disease in the 1960s. The disease has become a major economic restraint in highly affected areas such as Egypt and Israel, and is of constant concern in other counties. LWD causes dehydration and collapsing at a late stage of maize cultivation, starting from the male flowering phase. The disease causal agent, *Magnaporthiopsis maydis*, is a seed- and soil-borne phytoparasitic fungus, penetrating the roots at sprouting, colonizing the vascular system without external symptoms, and spreading upwards in the xylem, eventually blocking the water supply to the plant’s upperparts. Nowadays, the disease’s control relies mostly on identifying and developing resistant maize cultivars. Still, host resistance can be limited because *M. maydis* undergoes pathogenic variations, and virulent strains can eventually overcome the host immunity. This alarming status is driving researchers to continue to seek other control methods. The current review will summarize the various strategies tested over the years to minimize the disease damage. These options include agricultural (crop rotation, cover crop, no-till, flooding the land before sowing, and balanced soil fertility), physical (solar heating), allelochemical, biological, and chemical interventions. Some of these methods have shown promising success, while others have contributed to our understanding of the disease development and the environmental and host-related factors that have shaped its outcome. The most updated global knowledge about LWD control will be presented, and knowledge gaps and future aims will be discussed.

## 1. Introduction

*Zea mays* L. (maize, corn) is one of the world’s leading crops for food, feed, and fuel and as a raw material for different industrial products [1]. Worldwide annual maize production is expanding at a rate of 1.6%. It was forecast that this rate will not meet the global demand in 2050 [2]. Among many diseases threatening this cultivar [3,4], late wilt disease (LWD) has been reported so far in 10 countries and is considered a major concern in highly infected countries such as Egypt [5], Israel [6], India [7], Spain, and Portugal [8]. Economic losses due to LWD were up to 40% in Egypt [9], 50–100% in Israel [10,11], and 51% in India [12]. Incidences of the disease can reach 100% in Egypt and Israel, and 70% in India. Although the disease has not been reported in the United States, *M. maydis* is regarded as a potentially high-risk phytopathogen [13,14]. LWD harms yield production by erupting at the flowering growth phase, resulting in severe dehydration and plant death.

Since the discovery of LWD in Egypt in the early 1960s [15], worldwide scientific efforts have led to much progress in understanding the disease mode and the pathogen causing it, *Magnaporthiopsis maydis* [13]. Moreover, specific research tools for the study of LWD were developed and applied in the lab, in growth room experiments under controlled conditions, and in field trials. A significant part of these efforts was dedicated to creating diverse control methods to restrict the disease’s burst and spread and minimize its effect on commercial maize manufacture. Previously, we reviewed the techniques developed over the years to study LWD and monitor its causal agent [16]. A follow-up review summarized the accumulated scientific knowledge and future perspectives. These aspects include the geographic disease distribution, the pathogenesis (including the environmental factors affecting it), the symptoms’ evolvement, and their outcome effect on commercial production [17]. All the updated information regarding the pathogen itself, *M. maydis,* was also summarized. This includes the fungus life cycle, primary and secondary hosts, interactions with other phytopathogens, and the fungus’ ability to survive under different environmental conditions. These aspects will be summarized here briefly.

The current review focuses on the vast efforts dedicated in the past 60 years to late wilt disease control. The inspected control methods produced different degrees of success and include agricultural options (flood fallowing and balanced soil fertility) [18,19], biofriendly approaches [20], physical (solar heating) [21], allelochemical [22], and chemical pesticide [6,23,24] practices. Recently, the tillage system’s impact, the cover crop, and the crop rotation have been shown to serve as bioprotective factors against *M. maydis* [25,26].

A targeted research effort led to advancement in our capability to eradicate LWD chemically. A practical, efficient, and economic Azoxystrobin-based control protocol [10,24,27,28] was developed, which can be applied commercially to protect LWD-susceptible maize cultivars. Notwithstanding this recent encouraging achievement, the intensive chemical intervention has several short- and long-term drawbacks. In the short term, an intensive chemical application may cause the emergence of resistance to the fungicide. Such situations are becoming more and more common [29]. In the long term, phytoparasitic fungi chemical eradication may result in environmental, human, and animal hazards.

The limitation of chemical fungicides has become critical and is currently a global priority [30]. Hence, considerable research efforts in the past two decades were dedicated to seeking alternative methods of LWD control. Most of these efforts focus on eco-friendly substitutions to traditional chemical approaches. These consist of using *Trichoderma* spp. or other beneficial microorganisms as a biocontrol agent (see, for example, [20,31]). Late wilt green control studies also aimed at developing soil conservation practices that promote antagonizing mycorrhizal fungi (summarized by [31]). Even though this scientific course has been extensively explored against many harmful plant pathogens [32], in regard to LWD, substantial knowledge gaps exist. Consequently, the potential of green approaches to control *M. maydis* has only now been revealed.

Currently, the most eco-friendly, cost-effective, and efficient method to restrict *M. maydis* is by using highly resistant maize varieties [33,34]. Yet, the discovery of *M. maydis* highly aggressive isolates [8,35,36] is a constant problem. These fungal strains may threaten resistant maize cultivars. Indeed, growing resistance cultivars for extended periods in the same location may lead to gradual LWD susceptibility weakening [16,24]. This concerning situation pushes researchers to continue seeking new methods to control LWD.

The current review will summarize the many approaches tested to restrain the disease’s spread and damage, stating their advantages and limitations. It will also recommend the application of these methods in high- or low-risk scenarios (resulting from host resistance degree and soil infection load). Ultimately, the review will highlight knowledge gaps and future research focus points that should be addressed to advance LWD-safe commercial maize production capability.

## 2. Late Wilt Disease

### 2.1. The Pathogen

The late wilt causal agent, *M. maydis*, is a seed-borne and soil-borne vascular wilt fungal pathogen that penetrates the host roots and colonizes the xylem tissue [37,38]. The taxonomic tree of this fungus is: phylum: *Ascomycota*, subphylum: *Pezizomycotina*, class: *Sordariomycetes*, subclass: *Sordariomycetidae*, family: *Magnaporthaceae*, genus: *Magnaporthiopsis*, species: *Magnaporthiopsis maydis* (the most updated scientific name of the pathogen [39,40], Index Fungorum database, website: http://www.indexfungorum.org/names/NamesRecord.asp?RecordID=810225, accessed on date 30 November 2021). Former scientific names are *Cephalosporium maydis* (Samra, Sabet, & Hing, 1963) [41] and *Harpophora maydis* (Samra, Sabet, & Hing, 1963; Gams, 2000) [42].

To date, no perfect stage for *M. maydis* has been identified [43], and *M. maydis* reproduces asexually through sclerotia and spores [41]. This fungus may be considered necrotrophic because it thrives on the remains of dead plant tissues after killing its host. Yet, it is able to survive for a lengthy period (through the whole maize growth period) on living susceptible genotypes and asymptomatically in resistant genotypes and alternative species hosts. So, it may be better defined as a hemibiotrophic fungus.

*M. maydis* spread as sclerotia, spores, or hyphae on the plants’ residuals [38]. The pathogen can persist in the stubble and maize debris; no-till systems may help maintain it [13]. *M. maydis* can survive in the ground for lengthy periods or by thriving inside diverse host plants, such as lupine (*Lupinus termis* L.) [44], cotton (*Gossypium hirsutum* L.) [45,46], watermelon (*Citrullus lanatus*), and green foxtail (*Setaria viridis*) [45,47].

### 2.2. Geographic Distribution

LWD has been reported so far in 10 countries: Egypt (1961) [48], India (1970) [49], Hungary (1998) [50], Spain and Portugal (2011) [51], Israel (2013) [11], and possibly Nepal (2015) [52]. The report in Nepal referred to *Cephalosporium acremonium*, a synonym of the black bundle disease agent [38,53]. *Acremonium maydis*, the black bundle disease agent, had mistakenly been referred to as *M. maydis* in the past. There are also unconfirmed reports (summarized by Johal et al., 2004 [13]) that LWD was discovered in Italy, Romania, and Kenya.

Global LWD spreading was attributed mainly to infected seeds’ transmission. Indeed, Pecsi and Nemeth (1998) [50] presumed that late wilt spread to Hungary by importing infected seeds. *M. maydis* was detected in 39 out of 42 seed samples in Egypt [54]. In Hungary, Michail et al. (1999) [37] detected the fungus at a higher proportion in white maize cultivar seeds (1–9%) than in yellow cultivar seeds (1–3%). The pathogen was detected in the embryo, the endosperm, and the seed coat in 12 of the 13 seed samples tested.

### 2.3. Disease Cycle and Pathogenesis

The disease mode in LWD-sensitive maize cultivars is well detailed in the scientific literature. *M. maydis* infects maize seedlings mainly during the first three weeks by sowing through their roots or mesocotyl (the seed-coleoptile connecting tissue). As the plants grow, they are less infected and become LWD-resistant about 50 days after sowing [38]. The disease cycle starts when the fungus grows on the roots’ surface, producing hyphae with brown, short, thick-walled, and swollen cells [38]. After root penetration, *M. maydis* colonizes xylem tissue (identified 21 days after sowing) and is rapidly transferred to the upper parts of the plant. *M. maydis* may occasionally cause seed rot or pre-emergence damping-off under high inoculum pressure [55].

The second critical infection phase starts when tassels first emerge (ca. day 55–65, R1 silking, silks visible outside the husks). At this stage, the fungus hyphae and conidia appear throughout the stalk [38], pathogen DNA levels reach their highest point in the stems [11] (Figure 1), and the first aboveground symptoms are revealed. Later, when *M. maydis* colonizes the entire stalk, avascular tissue occlusion by hyphae and gum-like secreted materials occurs, resulting in water supply suffocation, rapid dehydration, and death [13,38]. At the end of the growing season (12–13 weeks after planting), the fungus is identified in different ear parts [37]. It is detected mainly in the ear branch but also in the cob, seeds, ear husks, and silk of naturally infected maize cultivars. The symptoms are intensified under drought stress [56,57]. Although the disease appears as patches scattered in the field in many cases [16], LWD may result in total field infection and total yield loss in heavily infected areas planted with susceptible maize cultivars [10,24]. A parallel asymptomatic infection mode, with some delay, occurs in resistant cultivars (Figure 1). This process can result in infected seeds that enhance the pathogen spread [10,11].

### 2.4. Disease Symptoms

The degree of disease symptoms is directly affected by soil infection load, maize genotype susceptibility, and environmental growing conditions, particularly the watering regime [18,38,56,57,58]. The first disease signs of LWD can be seen in the seedlings phase. *M. maydis* can destructively affect the seedlings’ aboveground emergence [55,59]. It may also cause reduced development of the sprouts’ roots [6,15], color alteration, and necrosis [22]. Small necrotic lesions (2–4 mm long) on the roots were documented three weeks post-inoculation [22]. Their size increased gradually to lengths of 10–14 mm. Intriguingly, similar dry dark-red local lesions on the roots near the soil surface were observed in *M. maydis*-inoculated cotton seedlings [46].

The first aboveground wilt symptoms appear approximately 50–60 days after sowing, near the flowering stage (from the R1 silking to the R2 blister) [38]. These first symptoms are parallel to the fungus’ establishment in the plant vascular system (see Section 2.3). They consist of rapid wilting of the first aboveground leaves that spread upwards during the subsequent two weeks [6]. As the disease advances, the leaves gradually alter their color and wilt [60]. Yellowish to brown-reddish streaks may be seen on the lower internode (Figure 2A–C) and a color alteration of the vascular bundles appears (Figure 2D). At this stage, the lower stem dehydrates (mainly at the internodes) and has a shrunken and hollow appearance (Figure 2E).

The roots and stalk vascular bundles become blocked, and their color in the stalk alters from light-yellow or white to dark-yellow and to brown [6,10,28,38]. Plants’ diseased leaves have a high proline content, probably linked with dryness stress due to limited water flow caused by tracheary elements plugging [61]. In addition, the infection may reduce the number of vascular bundles (seen in a cross-section of the internode). Late wilt infection is frequently associated with secondary pathogens infection that enhances the stem symptoms [62]. Eventually, fewer ears are produced, and if kernels do develop, they are often immature and damaged [6,11] (Figure 3) and infested with the pathogen. Seed quantity [63] and quality [28] are negatively correlated to disease severity. The infected kernels can result in seed rot and pre-emergence damping-off [64]. Finally, these disease processes can lead to the plant’s death (Figure 3).

## 3. Control Strategies

### 3.1. Host Resistance

The magnitude of LWD losses largely depends upon the degree of soil infestation and the susceptibility of the grown cultivars. The use of resistant genotypes is considered the best, most practical, eco-friendly, and cost-effective method of controlling the disease [65,66]. This method is preferred even though resistant hybrids to LWD are often low-yielding or have other undesirable agronomic characteristics [67]. A program to develop and identify new hybrid strains resistant to LWD has been operating in Egypt since the 1980s [68], in Israel for more than a decade (R&D North, MIGAL—Galilee Research Institute, Kiryat Shmona, Israel) [28] (Figure 4), and was also reported in India [7].

Significant efforts were directed towards using specific genetic markers for LWD to identify resistant germplasm and subsequently develop genetically resistant maize inbred lines [43]. Several studies suggest that many genes control LWD resistance in maize [70]. Yet, little scientific data exist regarding LWD resistance alongside high yield in maize [67]. LWD resistance inheritance is complex, having significant genotype × environment interactions. Hence, directed selection for LWD tolerance is probably less effective [71].

Nonetheless, DNA markers could be a suitable substitute for such traits in maize. To this end, identification and validation of closely linked markers are needed. Indeed, quantitative trait locus/loci (QTL) that confer LWD resistance have been identified and validated [71,72]. Further research investigation is necessary to detect stable QTL with considerable phenotypic variation explained [71].

Despite these meaningful efforts, the reasons for LWD susceptibility differences among maize cultivars remain obscure. It was gradually revealed that the infection process’ outcome results from chemical and histological differences between cultivars [73]. Phenols as phytochemical compounds have many functions; one is to protect plants against pathogens [74]. It was shown earlier that resistant maize plants contained higher total and free phenolic compounds than susceptible plants [75,76,77]. Phenolic acids (ferulic acids and cinnamic) found in vitro in resistant maize cultivars could suppress the growth of *M. maydis*. The phenolic content in all maize cultivars increased after infection. This increase was higher in resistant cultivars compared to susceptible cultivars. Lately, it was proven that under artificial infection stress, from 30 to 90 days after sowing (DAS), a linear increase in phenolic content was more pronounced in the resistant cultivar compared to the susceptible one [73].

In contrast, LWD-susceptible cultivars have a higher total soluble sugar content that may be favored by the pathogen as a carbon source of nutrition. The stalks’ sugar content generally decreased when the plants matured. Infection with *M. maydis* increased sugar content in both resistant and susceptible cultivars [78]. The tendency for the gradual increase in phenolic substance concentrations and the gradual decrease in sugar content in the plant tissues, along with the growth session, may be the reason for the plant’s acquired immunity to late wilt infection. Sabet et al. supported these findings and discovered that most plants were infected when the inoculum was applied to young plants. The *M. maydis* infection became less frequent until no maize plants were infected after seven weeks from sowing [38].

Furthermore, the pathogen was found in the resistant maize cultivars, mostly in the infected roots and rarely in the vessels of the stem’s lower internodes [79]. Laccase secretion in *M. maydis* was recently investigated in vitro, in response to different host tissues [80]. This enzyme can catalyze the oxidation of phenolic substrates and may act as a fungal defense against antifungal phenols secreted by the plant. Studying the laccase role in the pathogen’s interactions with diferent hosts would be very interesting and can contribute to our understanding of the resistance mechanism.

Particularly intriguing, in a field trial, in 72 old plants of a resistant cultivar, the *M. maydis* DNA levels in the roots, stems, and leaves were similar to the fungal DNA spreading in a susceptible cultivar on day 57 [11] (Figure 1). This may explain why resistance is sometimes limited to a specific growth period. Eventually, some maize genotypes that are non-symptomatic in the harvesting stage (typically the R3–R4 growth stages three to four weeks after fertilization) will dehydrate and die after an additional one or two weeks [5].

Another aspect that may contribute to the immunity of LWD tolerance by some maize genotypes is their tissue structure. Indeed, the roots of resistant cultivars had a different structure of the tissue surrounding the xylem vessels [76,77]. Resistant LWD cultivars, in comparison to susceptible ones, had increased thickness of sheath bundles surrounding the vascular bundles [73] and more layers of sclerenchymatous cells surrounding the xylem vessels. These tissue layers may act as mechanical barriers protecting bundles against pathogen invasion. Indeed, the fungus appears to penetrate directly through the host cells to ramify in the cortex. After that, the hypha progresses both in and between cells towards the xylem [38]. Hyphae aggregate at the endodermis prior to the fungus breaking into the vascular bundle, three weeks after sowing.

Vascular bundles of resistant inbred lines revealed a higher number of xylem vessels than susceptible ones [73]. The pith area was also much larger in resistant inbred lines than in susceptible ones. This supports the conclusion that occlusion plays a pivotal role in causing LWD symptoms [81]. Indeed, the *M. maydis* virulent isolates’ ability to plug the host plant xylem vessels was proven [73]. Thus, mechanical suppression of water uptake through such blocked vessels could be concluded. Supporting this conclusion are the growing pieces of evidence that high water potential reduces LWD damage in the field [18,82].

Finally, plant hormones may influence *M. maydis* pathogenesis, the severity of the disease symptoms, and host susceptibility [83]. It was shown that the plant growth hormones, auxin (indole-3-acetic acid), cytokinin (kinetin), and gibberellin (gibberellic acid), inhibit the development of the LWD pathogen in vitro [84] (Figure 5).

Despite auxin’s marked restricting effect on the pathogen’s growth in vitro, treatments based on dimethylamine salt of 2,4-D (dicots’ herbicide that mimics auxin influence, 96.9% active ingredient, Aminobar, Luxembourg Industries Ltd., Tel Aviv, Israel) were inefficient. Applying this compound in a plate assay (O. Degani, personal communication) and in the field using the driplines irrigation system [84] failed to suppress late wilt or prevent its symptoms. So, do plant hormones play a vital role in the tolerance mechanism in some maize cultivars? This question should be explored in future studies.

Alongside these efforts, the development of new methods for tracking the pathogen [7,85,86], estimating its distribution and damage [8,14,87], and controlling it in various ways [22,25,65,88,89] remain major goals.

### 3.2. Chemical Control

#### 3.2.1. From In Vitro Evaluation to a Field Assay of Selected Fungicides

It is important to locate LWD antagonists’ fungicides while evaluating their phytotoxicity and efficiency against *M. maydis* and other associated fungi (the stalk-rot complex) involved in LWD [90]. Together with this effort, developing rapid and efficient screening approaches to assess the potential of these fungicides is needed [10,91]. Preliminary tests on growth media plates aimed at screening many chemical preparations rapidly and indicating their efficiency can be conducted with minimal investment and can save time and effort. After eliminating inadequate potential fungicides, selected compounds will be verified in seeds, detached roots, and sprouts’ assays. Only in the final phase will a limited number of high-potential selected pesticides be tested in a field experiment throughout the growth season (Figure 6). While this method is important to reduce the high investment and the long time required for field trials, it is not flawless.

It is necessary to acknowledge that culture plate screening, seeds’ and detached roots’ assay, and even potted young plant trials under controlled conditions have partial ability to forecast field results [10]. Nonetheless, potted seedling experiments for chosen fungicides are essential. Such experiments do not depend on the year or season required for field growth and can be carried out throughout the year. They are also not affected by environmental high variability conditions accompanying open-air experiments.

#### 3.2.2. Seed Coating

Seed coating is a common technique to protect emerging maize sprouts from soil-borne fungal pathogens. It has the potential of playing a vital role in shielding against *M. maydis*. Since the initial infection occurs during the seedling development period [38], the LWD can be managed efficiently with a fungicide seed treatment. Indeed, such attempts have been made in the past. In India, Begum et al. (1989, 1996) showed that seed treatments with captan, carboxin, carbendazim, and thiram significantly reduced late wilt severity and increased yield in the field [92,93]. Yield increased by 25.9% for carbendazim and 34.1% for captan (at 1 g/kg seed).

In contrast, seed treatments failed to prevent late wilt in Egyptian trials [23,94]. According to Johal et al. (2014) [13], such differences can result from variances in the virulence, chemical sensitivity of *M. maydis* isolates, or the consequences of the stalk-rot disease complex in Egyptian soils [90]. Nonetheless, non-chemical seed treatments were tested successfully in Egypt [95]. It was demonstrated that the usage of some rhizobacterial strains (*Bacillus subtilis* and *Pseudomonas fluorescens*) and organic compounds, such as compost and humic substances, as a seed treatment can prevent maize infection with LWD. In addition, under field conditions in regular and saline soil, rhizobacterial seed coating with *B. subtilis* and *P. fluorescens* either alone or in combination significantly decreased disease infection and enhanced plant yield compared to the control [88].

In Israel, in relatively resistant maize with only minor LWD symptoms, the Azoxystrobin seed coating (0.0025 mg active ingredient/seed) prevented fungal development and increased plant and cob weight [10]. Yet, this treatment could not defend a susceptible maize cultivar in heavily infested soil at the disease’s wilting burst (60 DAS) and later on. In follow-up work [28], the Azoxystrobin + Difenoconazole seed coating (AS + DC, “Ortiva-Top”, manufactured by Syngenta, Basel, Switzerland, supplied by Adama Makhteshim, Airport City, Israel) was applied. This treatment was efficient in the initial growth stages (up to 50 DAS). At the later growth stages, the AS-DC seed coating provided an additional layer of protection when combined with the injection of fungicides into the irrigation system, as elaborated in the following Section 3.2.3. This combination significantly reduced disease symptoms and the pathogen DNA in the plants’ lower stem to near-zero levels, resulting in high yield (quantity and quality) production.

#### 3.2.3. Soil Treatments

Systemic fungicides and their fungi-toxic products translocate to maize leaves within two days and can last in maize roots for 90 days [23], so *M. maydis* may be repressed by these chemicals within the root, as well as in the soil. Thus, soil treatments with systemic fungicides, such as Benomyl (Benlate, methyl 1-(Butylcarbamoy l)-1H-1,3-benzimidazol-2-yl methylcarbamate, CAS no. 17804-35-2), were the preferred method of dealing with the disease.

In pot experiments with 15 fungicides conducted by Singh and Siradhana (1989) [23,96], LWD was significantly reduced by soil applications of Benomyl and Bavistin (CAS no. 63278-70-6, carbendazim), both at 0.1% and by 2% Bayleton (CAS no. 43121-43-3, triadimefon). Yet, over the years, during LWD research, it has become clear that fungicides that were successfully proven in pots may fail in field experiments. An example is the fungicide Benomyl, introduced in 1968 by DuPont (Wilmington, DE, USA). It is a systemic benzimidazole that is selectively toxic to microorganisms and invertebrates, especially earthworms, but non-toxic to mammals. It was found that Benomyl at 2.5 to 100 ppm concentrations completely inhibits LWD in pots (but not by applications 30 days after sowing or by seed treatments) [96]. Still, the application of 10 kg Benomyl/4200 m^2^ in the field failed to prevent the disease [94]. Lack of success has been attributed to reduced absorption of systemic fungicides by maize in the field soil compared to pot experiments [94].

The application method can be game-changing in this regard, as was proven in Egypt [23] and Israel (see Section 3.2.4). Abd-el-Rahim and colleagues (1982) found that the systemic fungicide Benylate applied at four 15-day intervals (2.5 kg/acre) after sowing resulted in the best LWD control [13]. In Israel, the application of fungicides at 15-day time intervals from sowing is an obligatory condition to achieve effective LWD control [24,27,28], as discussed in detail below. Nevertheless, it was estimated that the cost and labor required for frequent fungicide applications in the United States make this control method prohibitively expensive [13].

#### 3.2.4. Azoxystrobin Irrigation-Based Treatments

Several chemicals were examined for their ability to restrict *M. maydis*, the causal disease agent in recent years [24,28]. Fast plate assay proved the success of Azoxystrobin against the pathogen (Figure 6), but applying its based commercial preparation in the field by spraying was ineffective [24]. In contrast, injecting Azoxystrobin directly into a dripline assigned for each row 15, 30, and 45 DAS inhibited wilt symptom development and recovered cob yield by 100%. However, this method is not always feasible, and to the best of our knowledge, has never been tested commercially due to the high cost of the irrigation system.

More recently, an efficient and more economically applicable solution to LWD was suggested that could be applied on a large scale to shield-susceptible corn varieties in commercial fields [28]. This application is based on antifungal mixtures having a different mode of action to prevent resistance development. The method involved seed coating and injection of Azoxystrobin and Difenoconazole mixture (AS+DC) into the irrigation system at three 15-day intervals from sowing. Economic efficacy was reached using one dripline for two adjacent rows (a row spacing of 50 cm instead of 96 cm). The short row spacing guaranteed the effective concentration of the antifungal compound near the roots. A quantitative real-time PCR (qPCR)-based molecular detection method, proven earlier [10], showed that due to the AS + DC treatment, *M. maydis* DNA levels in the host tissue dropped to near-zero values. In the coupled-row-based cultivation, this treatment reduced wilt symptoms by 41%. At the same time, it recovered yield to the common level in healthy fields (1.6 times more than the non-protected control). In addition, yield quality (A-class cobs at a weight exceeding 250 g) increased from 58% to 75% in this treatment.

Nowadays, drip irrigation is considered one of the most effective chemical methods to restrict maize LWD but it is the most expensive of the present alternatives [28]. The value of drip extension lines (without installation) is 50–60% of total costs, so reducing the extension number by half could save about 30–40% of the irrigation expenses and lead to savings in labor and time. Consequently, this successful treatment is now more cost effective and can be applied commercially to protect susceptible maize hybrids in infested fields against late wilt.

#### 3.2.5. Fungicide Resistance

Azoxystrobin, a member of the Qo-inhibiting fungicides (QoIs) class, is considered one of the most important agricultural antifungals [97]. However, growing resistance to these antifungals and the resulting control failure are become a significant problem (see, for example, such cases in *Magnaporthe grisea* [98]). In the past two decades, ca. 30 phytopathogen species distributed across 20 genera were reported to exhibit field resistance toward QoI fungicides [99]. QoI fungicides disrupt respiration in the mitochondria by binding to the cytochrome bc1 enzyme complex Qo site, impairing electron transfer and thus inhibiting mitochondrial respiration and ATP production [97]. Since their mode of action is based on a single site, QoI fungicides pose a high risk for revealing fungicide resistance. Indeed, Azoxystrobin targets site mutations in cyt b gene (G143A, F129L) and additional mechanisms have been reported [100].

Under such circumstances, the suggestion is to avoid using QoI antifungals or to apply anti-resistance-emergence strategies such as mixing low-risk and high-risk antifungal compounds. These steps may help delay the emergence of resistance to the high-risk fungicide [101]. In addition, cross-resistance has been documented among all members of the QoI group (Fungicide Resistance Action Committee (FRAC) Code List© 2021). Consequently, mixing Azoxystrobin with other antifungal agents is critical for LWD control.

### 3.3. Biological Control

#### 3.3.1. Strengthening Beneficial Microorganism Communities in the Soil and Their Secreted Metabolites

Since fungicide treatment limitation exerts increasing pressure in many countries due to environmental and potential health risks, searching for alternatives to cope with LWD is a continuous effort. Biopesticides are environmentally friendly. Therefore, this approach occupies an increasingly central place in worldwide scientific research to this end. To address this challenge, many studies were directed towards LWD biological control [20,31,102,103,104]. These methods include operating and strengthening beneficial microorganism communities in the soil (for example, by compost addition [102]) or direct intervention using antagonistic bacteria and fungi or their secreted metabolites. An example is *B. subtilis* MF497446 and *Pseudomonas koreensis* (two plant-growth-promoting rhizobacteria [31]). This microorganism’s mixture resulted in siderophore production and *M. maydis* repression. Moreover, the treatment prevented pre- and post-emergence damping-off and promoted the growth of greenhouse plants, proving to be highly efficient in the field in reducing infection and increasing the yield [31].

Seed treatments with biocontrol formulations (*B. subtilis*, *Bacillus pumilus*, *P. fluorescens*, *Epicoccum nigrum*) were suggested for maize LWD control and tested in the field with encouraging results [89]. The seed treatments were tested in two growing seasons, reducing the pathogen impact on pre-emergence damping-off, wilt incidences, and crop yield. In an additional study, the influence of maize root colonization by microorganisms was also practical for this purpose [104]. In this study, the rhizosphere actinomycetes (*Streptomyces rochei*, *S. graminofaciens*, *S. gibsonii*, *S. annulatus*, *Candida maltosa*, *C. glabrata*, *C. slooffii*) and the fungi *Rhodotorula rubra* significantly reduced *M. maydis* growth in vitro. These treatments were also highly influential in seed dressing under greenhouse-controlled conditions. Adding these species in the absence of the LWD pathogen significantly increased maize plant growth parameters [104].

Another example is the use of a filtrate of mixed strains of the cyanobacteria *Anabaena oryzae*, *Nostocmuscorum*, and *N. calcicolawere* [102]. An alternative approach was to use marine algae and cyanobacteria *A**. oryzae* extracts exhibiting antifungal activities to target the LWD pathogen [103]. These algae include *Corallina elongata*, *Jania rubens*, *Laurencia obtusa*, *Gelidium crinale*, *Enteromorpha compressa*, and *Ulva fasciata*. The results are encouraging, with *U. fasciata* and *A. oryzae* exhibiting high antifungal activity against *M. maydis* [103].

#### 3.3.2. *Trichoderma* spp. Maize Late Wilt Biocontrol

Late wilt disease can also be biologically controlled using *Trichoderma* spp. This genus’ species can form endophytic mutualistic relationships with various plant species [105]. Other *Trichoderma* species have been identified to possess biocontrol potential against plants’ fungal pathogens [106]. For instance, *Trichoderma cutaneum* reduced the incidence of LWD of maize under greenhouse conditions by 89% compared to the non-protected control (from 94% to 11%) [104]. Likewise, *Trichoderma harzianum* treatment in the field reduced the pre-emergence damping-off from 47% to 32% and increased the plants’ survival by 59% [64]. The application of *Trichoderma viride* alone, or even better with chitosan NPs combined with the mycorrhizae *Glomus mosseae*, controlled late wilt in the greenhouse and in field trials, and enhanced the plants’ growth indices [107]. Using the same combination of *T. viride* and mycorrhizae without chitosan NPs led to a low-efficiency LWD control. To maximize the impact of *Trichoderma*-based treatments, Elshahawy and El-Sayed (2018) [20] showed that extracts of the microalgae, *Chlorella vulgaris*, with each of the *Trichoderma* species, *T. virens* and *Trichoderma koningii*, led to effective LWD control in the greenhouse and in the field. These combinations led to a 72% reduction in disease frequency in the greenhouse and a 2.5-fold higher grain yield in the field [20].

The potential for using *Trichoderma*-based treatment against Israeli *M. maydis* strains has only recently been tested [108]. Examining eight marine [109] and soil isolates of *Trichoderma* spp. known for their high mycoparasitic potential revealed that *Trichoderma longibrachiatum* (T7407) and *Trichoderma asperelloides* (T203) isolates have solid antagonistic activity against the Israeli *M maydis* strain. These eco-friendly agents were tested in a series of experiments in the laboratory (Figure 7) and in a growth room under controlled conditions until their final examination in pots under field conditions throughout an entire growing season [108]. The *T. longibrachiatum* (T7407) green treatment significantly improved growth and yield indices to healthy plants’ levels, reduced pathogen DNA in the plants’ tissues by 98%, and prevented disease symptoms (Figure 8).

In solid and submerged media culture growth assays, these species secrete soluble metabolites that inhibit or kill the maize pathogen [108]. Such a metabolite was recently isolated and identified as pyrone 6-pentyl-2H-pyran-2-one (6-pentyl-α-pyrone or 6-PP) [110]. This potent *M. maydis* antifungal compound is secreted by *Trichoderma asperellum* (P1), an endophyte separated in our laboratory from maize seeds of a cultivar susceptible to LWD [111]. The 6-PP metabolite was previously identified as one of the key bioactive compounds of several *Trichoderma* species [112].

To test the bioprotective potential of these isolates for commercial maize production, follow-up work was performed. *T. asperellum* (P1) significantly improved the infected sprouts’ growth parameters and reduced *M. maydis* DNA in their roots [111]. The application of this endophytic species also excels in the field over an entire growing season [58]. At the season’s end, the *T. asperellum* treatment resulted in 1.6- and 1.3-fold improvement in the LWD symptoms in the lower stem and cob, respectively. Furthermore, this treatment led to 4.9-fold lower *M. maydis* DNA levels in the plants.

It was reported that the fungus *T. longibrachiatum* (T7407) produced 29% more fungal inhibitory effects than *T. asperellum* (P1) in solid growth media [110]. Active ingredients that will be isolated from T7407 could probably be more potent. The active ingredient isolation and identification in this and other potential *Trichoderma* spp.-secreted antifungal metabolites are of great importance and should be the focus of follow-up works.

#### 3.3.3. Manipulating the Plant Microbiome

The natural microorganisms’ communities inhabiting the plant’s phyllosphere (the plant’s aboveground habitat) or the rhizosphere (the roots’ nearby habitat) comprise pathogenic and non-pathogenic members that may protect the plant from pathogens. The soil pathogen, *M. maydis*, interacts with the maize endophytes, which may provide the plant’s first defense line. Recently, such endophytes were isolated from six sweet and fodder maize hybrids with deferent sensitivity to LWD [111]. These include ten fungal species and one bacterial species, *B. subtilis*. The fungal species belonged to *Chaetomium*, *Trichoderma*, *Penicillium*, *Rhizopus*, *Alternaria*, and *Fusarium* genera. Some are plant pathogens (*Alternaria alternata* and *Fusarium proliferatum*), so they play complex roles in their interactions with the host plant. Enriching seeds with either *Chaetomium subaffine* or *T. asper**ellum* significantly promoted the infected plants’ growth parameters 42 days past sowing. The fungal species *Chaetomium cochliodes*, *T. asperellum,*
*Penicillium citrinum*, and the bacteria *B. subtilis* treatments reduced the LWD pathogen DNA in the host plant’s roots [111].

In cotton plants, interactions between *M. maydis* and *F. oxysporum* (the cotton wilt agent) led to an interesting result—reduced symptoms of the cotton wilt disease [46]. The infection reduction was maximized when *M. maydis* preceded *F. oxysporum* in the soil compared to simultaneous inoculation. Co-infection of the plants with both fungi resulted in immunity compared to infection with *F. oxysporum* alone. In contrast, there was little or no protective outcome when *M. maydis* was added to the soil after *F. oxysporum* [46].

Similar antagonistic relationships in cotton plants were found between *M. maydis* and *Macrophomina phaseolina*, the charcoal rot disease agent [45]. In a plate confrontation assay, the fungi formed a clear line between the colonies at the meeting area, representing strong mutual antagonism. Under field conditions, *M. phaseolina* markedly recovered the plants’ health while eliminating the severe late wilt symptoms [45].

### 3.4. Agrotechnical Measures

#### 3.4.1. Various Cultural Methods

Agrotechnical applications were also reported in varying degrees of maturity in field experiments. These methods, which had a beneficial impact on LWD suppression, include excessive irrigation (pot experiments conducted in an open-air enclosure) [56] and applying plant extracts (*Lycium europaeum* [22], aloe vera (*Barbados Aloe*) fleshy leaves, onion bulbs, garlic cloves, jimsonweed, and peppermint leaves [113]). In addition, soil solarization to increase temperatures above 35 °C with transparent polyethylene film [21], balanced soil fertility [19,114], and avoiding drought stress [56,61] can reduce LWD severity and yield losses. *Magnaporthiopsis maydis* survival is restricted to the top 20 cm of soil [115], and survival depends mainly on infected crop residues. Thus, sanitation measures such as deep tillage and annual plowing may significantly impact LWD (Dr. J. Leslie, personal communication) [14]. The use of no-till maize systems could eventually result in increased virulence of the pathogen or inoculum build-up in the soil. Finally, it was shown that organic compounds (compost tea, olive mill wastewater, and humic acid as seed treatment) [95] and non-traditional methods such as adding nanosilica and zinc oxide nanoparticles [89] have promising potential to reduce late wilt in the field.

#### 3.4.2. Beneficial Mycorrhizal Communities

Preserving soil mycorrhizal fungi between growth periods has been essential in crop protection (summarized by [31]). Results from other crop diseases suggest that under no-till cropping, crops in a rotation or selected cover crops could support building mycorrhizal communities that function through a sequence of several crops [116]. Intensive tillage combined with extended periods where the field is unprocessed results in damaging the integrity of mycorrhizal networks. Preserving the integrity and continuity of the soil mycorrhizal networks may provide the plant with higher resistance to soil diseases [117], including late wilt disease [25]. Indeed, arbuscular mycorrhizal fungi (AMF) can improve plants’ resistance to biotic and abiotic stresses. This results from activating the plant’s local and systemic defense mechanisms [118].

Although global scientific efforts are focusing on seeking solutions to LWD based on eco-friendly biological methods, a lack of information exists on maize performance under LWD stress in reduced tillage and crop rotation. In Egypt, crop rotation with rice (*Oryza sativa*) provides some control [119]. Maize did not develop late wilt following paddy-cultivated rice, which increases the availability of Mn for subsequent crops [60]. In India, *M. maydis* remained viable in stem residues on the soil’s surface for one year, but the pathogen could not be recovered after ten months from stem pieces buried at 10 cm [82]. High soil moisture enhances LWD development, but saturated soil reduces it [119]. In Portugal, Patanita et al. (2020) [25] showed that *M. maydis* presence and grain production were significantly reduced when both minimum tillage and cover crop were applied. It was also found that arbuscular root colonization was higher following these practices.

In Israel, agricultural practice based on conserving soil microflora integrity (by avoiding tillage) and influencing its nature (by cultivating specific crops in a dual-season growth) was applied [26]. When maize was seeded on wheat soil, a significant improvement in the shoot’s fresh weight (47–54%) and cob (36–46%) was achieved compared to the other treatments (clover soil, commercial mycorrhiza preparation, and the control). This achievement was not affected drastically by tillage. It was followed by a sharp decrease in disease symptoms (73%) and the pathogen’s presence (82–64%) in the plants’ tissues. It was concluded that since wheat and maize are more closely related (they are both *Poaceae*) than clover and maize, they might share similar mycorrhizal networks adapted to perform better with these crops. Indeed, the growth promotion mentioned above and LWD tolerance were reduced in the clover-maize sequence, even more so when tillage was applied. To support this hypothesis, it was found that crop plants acquired a mycorrhizal community closely related to that of the former host plant and different from that found when the soil was disturbed by tillage or not cropped before the growth [116].

With the novelty of such studies, it is essential to indicate that crop rotation may have a long-term impact on soil fungus populations, which may only be evident after an extended period. In addition, no single cropping system is preferred for all fungi [120]. Thus, a tailored solution should be planned wisely to address the cultivar that needs to be protected, the crops in the rotation, the tillage system, and the pathogen/disease stresses. Such situations should be inspected in subsequent studies. Closely related crops such as maize and wheat may benefit from the same control strategy.

## 4. Future Challenges and Opportunities

Late wilt disease of maize is a challenging disease that imposes a significant economic price in infected areas. The primary method of LWD restriction, the use of resistance germline, is an efficient, economical, and environmentally friendly solution that should be maintained and improved. Breeding for resistance is a continuous effort and is becoming especially essential when new virulent isolates of *M. maydis* are recognized [13]. However, limited information is available on the inheritance of resistance. Sources of maize genetic resistance must be pursued, the resistance mechanisms should be explored and clarified, and methods to rapidly identify new and stable LWD resistance maize cultivars should be developed (or those already set could be improved). The development of specific genetic markers for resistance to late wilt would greatly facilitate the incorporation of resistance into adapted hybrids [14].

Alongside this primary purpose, the continued development of other solutions to efficiently control the LWD pathogen, *M. maydis*, for commercial maize production is urgently needed in Israel, Egypt, Spain, India, and other countries [7,8,59,121]. Chemical control of LWD is considered the most effective solution in highly infected areas to protect susceptible maize cultivars. In recent years, research efforts devoted to identifying and applying chemical pesticides against LWD in Israel produced encouraging results [24,27,28]. Azoxystrobin-based commercial mixtures can be applied at a timetable adjusted to *M. maydis* pathogenesis and shield-susceptible maize cultivars, even in a highly infected area. Yet, such a method is dependent on dripline irrigation that is not suitable in many maize-growing areas and is less economical. In addition, intensive chemical treatment has several drawbacks in the short and long term. In the short term, an intensive chemical application may result in the appearance of fungicide resistance. Such cases have gradually become more common [29]. Fungicides that restrain plant parasitic fungi may lead to human, animal, and environmental risks in the long term.

Thus, future efforts devoted to this direction should focus on searching for new, hazard-free chemicals that are highly effective against *M. maydis*. Varied, effective application techniques of successfully tested pesticides should be developed to meet the growers’ needs. There is also a strong necessity to minimize the pesticides’ dosages and to produce and inspect blends of fungicides to prevent the emergence of fungal resistance lines.

One option that could solve many of the problems posed by chemical crop protection is combining chemical and biological approaches [30]. This solution has been proposed to reduce fungicide doses (and their residues’ impact on harvested crops). In addition, combining antifungal chemical and biological treatments reduces the selection pressure on pathogens and thereby the chances of resistance development. However, to enable this method of implementation, many knowledge gaps need to be addressed.

Today, biopesticides have gained considerable attention in the scientific world because they are a vital alternative to traditional chemical pesticides used to protect field crops. While the development of new eco-friendly options is at the forefront of many novel research studies, there is still a requirement for improved existing protocols, as demonstrated by Elshahawy and El-Sayed, 2018 [20]. Their work suggests maximizing the efficacy of *Trichoderma* against *M. maydis* using freshwater microalgae extracts. Such a research direction is opening the door for many similar solutions that, if adequately developed and tested, may produce highly effective and economical solutions to LWD.

Bio-friendly protective microorganisms produce secondary metabolites with strong antifungal activity. Such a metabolite is 6-pentyl-α-pyrone (6-PP) [110]. This ingredient plays a pivotal role in the biological control of several important phytopathogens and thus may provide a broad defense to the plant. The clean *T. asperellum* 6-PP-secreted product has a highly effective antifungal activity against *M. maydis*. Such an isolation and identification process is the first step in discovering its commercial potential as a new fungicide. Future studies should confirm this purified component’s effectiveness as a seed coating or other preventive treatments to shield highly susceptible maize hybrids against LWD.

Any in vitro work to identify new metabolites with *M. maydis* antagonism must be followed by in vivo work, which would include *M. maydis*-infected seedlings and mature plants over an entire growth period under field conditions. These further steps, the sprouts, and the field trials will ultimately enable reaching a concluding decision—will the metabolite be practical on a commercial field scale? If so, an identified and validated chemical or mixture of a few materials having high efficiency against the LWD causal agent could be proposed as a new pesticide’s main ingredient and commercially developed into a future product.

As recently shown, one important source of potential microorganisms exhibiting powerful activity against *M. maydis* is the maize plants themselves [111]. The roots or seeds of maize plants (apparently LWD-susceptible cultivars are preferred) are inhabited by many beneficial fungi and bacteria that shield the plant from outside invading pathogens. Identification of these members of the plant microbiome and exploring their potential may open a vast array of new possibilities to control *M. maydis*. So, we are encouraged to widen and deepen our understanding to reveal the true potential of the maize microbiome in the plant survival struggle against the pathogen. A better understanding of these interactions under natural conditions will help us understand, influence, and take advantage of the endophyte-based biocontrol potential.

## Figures and Tables

**Figure 1 pathogens-11-00013-f001:**
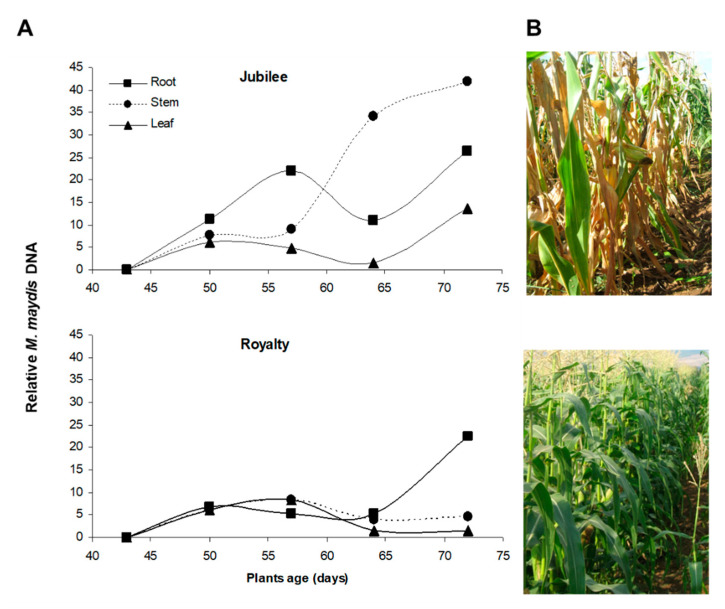
Molecular diagnostic of *Magnaporthiopsis maydis* pathogenesis in susceptible and resistant maize (*Zea mays* L.) cultivars. The experiment is described in [11]. (**A**) The progression of late wilt disease (LWD) from 43 days after seeding until maturity was estimated using a PCR-based method. The LWD-sensitive Jubilee cv. (upper panel) and the relatively LWD-resistant Royalty cv. (lower panel) were inspected (both sweet maize cultivars from Pop Vriend Seeds B.V., Andijk, The Netherlands, supplied by Eden Seeds, Reut, Israel). A semi-quantitative analysis of DNA isolated from the maize plants’ roots, stems, and leaves was performed weekly. The optical density of the *M. maydis*-specific primer bands was measured and normalized against the optical density of the control—ribosomal DNA (rDNA) bands. (**B**) A representative photograph of Jubilee cv. (upper panel) and Royalty (lower panel) plants on day 76 of growth.

**Figure 2 pathogens-11-00013-f002:**
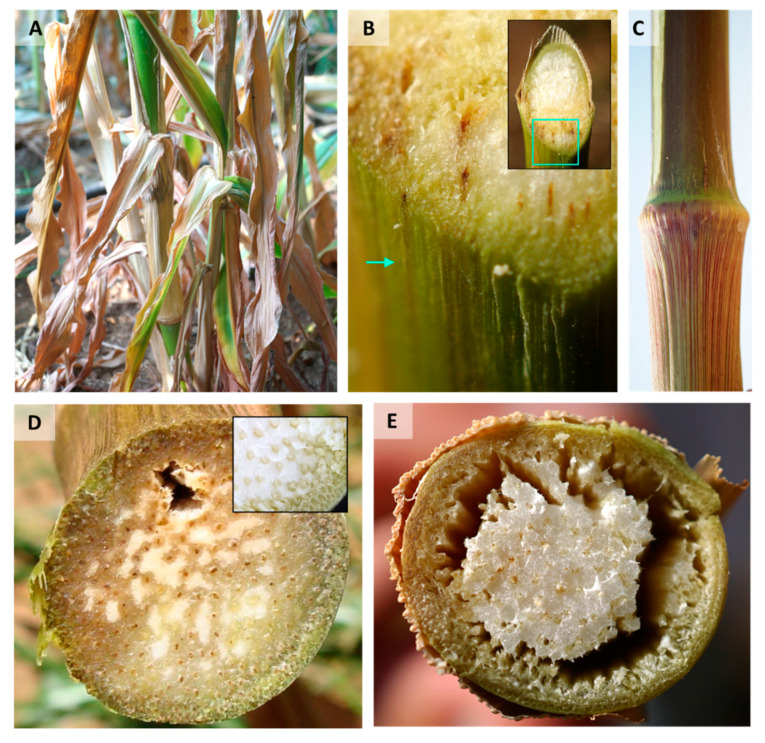
Lower stem symptoms of late wilt disease (LWD) of maize in a susceptible maize cultivar. All photos were taken of the lower stems (near the first internode) at harvest (ca. 80 days from sowing). (**A**) Wilt symptoms of lower plants’ leaves and stems in the field. (**B**) Cross-section magnification showing the vascular bundles’ color alteration to red-brown. Insert: cross-section in which a green frame marks the magnification area. The green arrow points to the surface color change. (**C**) The color alteration of the vascular bundles appears as red strips on the lower stem surface. (**D**) Cross-section of field LWD-infected plants. Late wilt diseased plants have a tissue color alteration to a yellow-brown hue and vascular bundle occlusion (adapted from [16]). Insert: photo taken from a healthy plant. (**E**) Cross-section of a severe late wilt diseased plant showing the breakdown of the parenchyma tissue.

**Figure 3 pathogens-11-00013-f003:**
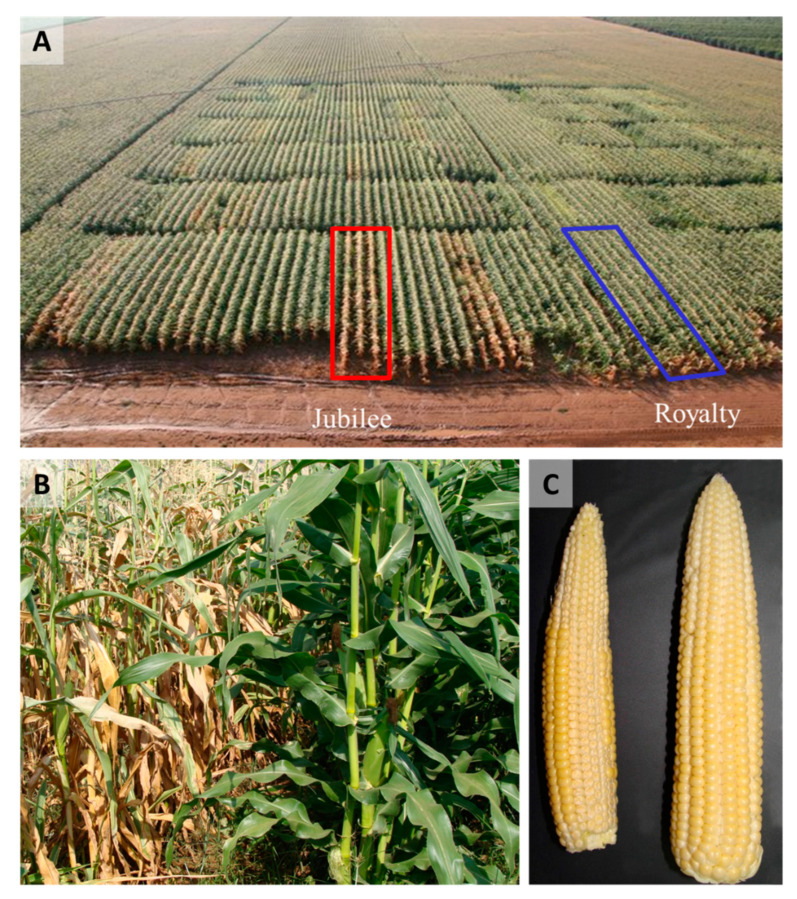
Maize late wilt disease (LWD) symptoms in the field 75 days after sowing (adapted from [6,11]). An observation for assessing cultivar resistance to *Magnaporthiopsis maydis* infection was conducted in a sweet corn field in the Hula Valley (Neot Mordechai, Upper Galilee, northern Israel). LWD-sensitive Jubilee cv. and relatively LWD-resistant Royalty cv. are shown (see Figure 1). (**A**) Aerial photograph (taken by Asaf Solomon) of the maize field, in which the two cultivars are marked: wilted Jubilee cv. plants (red box) and healthy Royalty cv. plants (blue box). (**B**) Ground photograph of the two representative cultivars: Jubilee (**left**) and Royalty (**right**). (**C**) Cob samples of the cultivars tested: Jubilee (**left**) and Royalty (**right**).

**Figure 4 pathogens-11-00013-f004:**
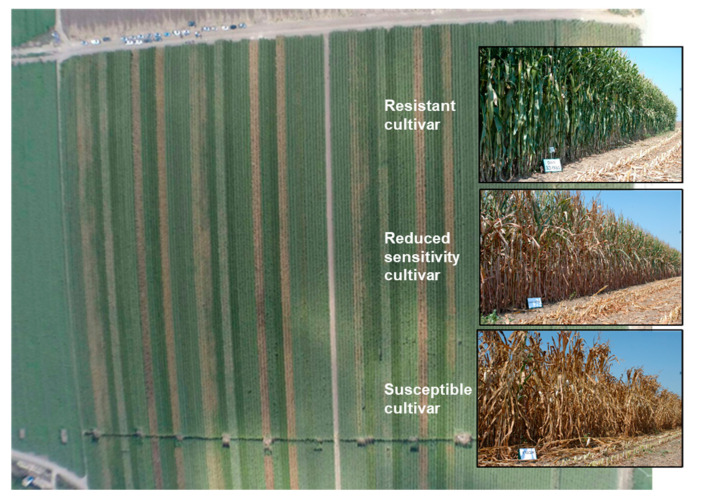
Cultivars’ resistance test for late wilt disease (adapted from [69]). A semi-commercial examination of fodder corn genotypes was conducted in the southern coastal plain of Israel (Yavne field) in 2014. The experiment included 14 maize cultivars in 3 repetitions, with each plot containing 6 rows measuring about 200 m in length. The experimental area photo was taken close to harvesting (day 99 from sowing) by David Katsav. The brown lines are late wilt diseased cultivars with severe dehydration, while the green lines are healthy cultivars. The photos on the right: resistance cultivar—Pan 33–031 (Eden Seeds, Hatzav, Israel); reduced sensitivity cultivar—Colossus from HSR Seeds (CTS, Hod Hasharon, Israel); susceptible cultivar—Avgaro (Hazera Seeds Ltd., Berurim MP Shikmim, Israel).

**Figure 5 pathogens-11-00013-f005:**
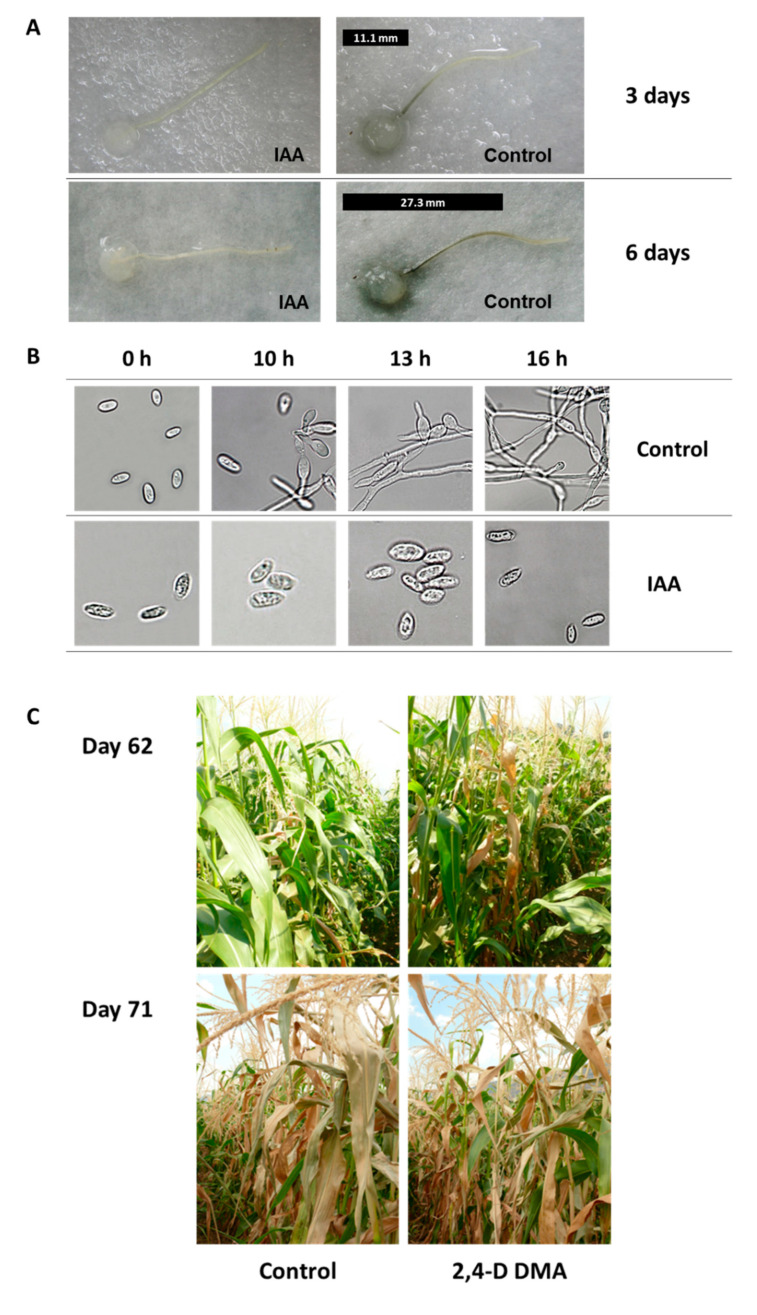
The impact of auxin (hormone indole-3-acetic acid, IAA) on *Magnaporthiopsis maydis* growth in vitro and in the field (adapted from [84]). (**A**) Detached root pathogenicity assay for the influence of auxin. Young, white side-roots, ca. 2 cm long, were cut from 20-day-old potted maize (*Zea mays* L., Jubilee cv.) seedlings and inoculated by placing a 6 mm diameter *M. maydis* culture agar disk taken from the margins of a 4–6-day-old colony (grown at 28 ± 1 °C in the dark) on each root’s cut end. The inoculated roots were positioned separately in Petri dishes that contain auxin (IAA, 100 mg/L) or distilled and deionized water, DDW (control). The palates were incubated at 28 ± 1 °C in the dark. Development of *M. maydis* infection thread (a dark filament) inside the roots was qualitatively evaluated after three and six days. The infection thread length is marked in the photo by a black line placed near each root. (**B**) Effect of plant hormones on *M. maydis* spore germination. Spores were washed from the surface of 4-day-old colonies and suspended in watery solutions containing IAA. The spores’ suspensions were then incubated at 28 ± 1 °C in the dark, in a rotary shaker (at 150 rpm) for the indicated times. The length of each spore is ca. 10 µm. (**C**) Field experiments to assess the efficiency of dimethylamine salt of 2,4-dichlorophenoxyacetic acid (2,4-D DMA, mimics auxin’s influence) in controlling late wilt. The experiment was conducted in an infested sweet corn field in the Hula Valley (upper Galilee, northern Israel). The Jubilee cv. plants were treated separately with 2,4-D DMA (96.9% active ingredient, Aminobar, Luxembourg Industries Ltd., Israel). The treatment was applied at a dosage of 150 cm^3^/0.1 ha, 15, 30, and 45 days from sowing. The treated and control (untreated) plants were photographed 62 and 71 days past sowing.

**Figure 6 pathogens-11-00013-f006:**
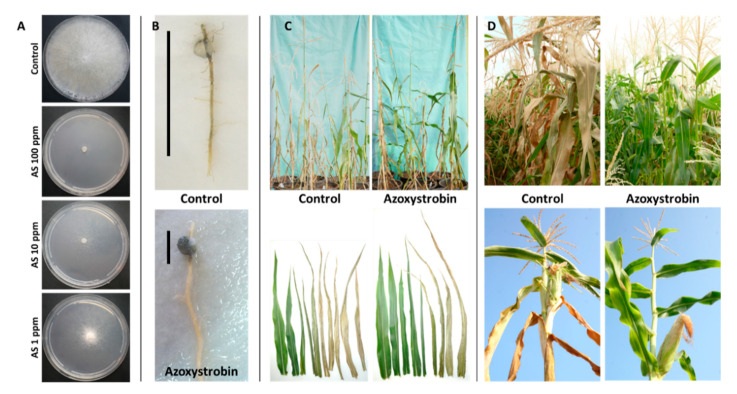
From in vitro evaluation to a field assay of Azoxystrobin. (**A**) Agar plate assay to evaluate the inhibition of *Magnaporthiopsis maydis* mycelial growth by Azoxystrobin (AS, CAS no. 131860-33-8, Amistar, Makhteshim Agan, Airport City, Israel) (adapted from [6]). Photos were taken five days after incubation at 28 °C in the dark. The fungicide was evaluated at a rate of 1, 10, and 100 mg/L active ingredients. Control—potato dextrose agar (PDA) plate without fungicide. (**B**) Detached root pathogenicity assay (adapted from [6]). The main (longest) inner feeder root was cut from a potted 3-week-old Jubilee cv. maize sprout. The detached root was inoculated by placing an *M. maydis* culture disk (6 mm diameter) taken from the margins of a 4–6-day-old colony, two cm away from its cut end. The inoculated roots were incubated in a moist atmosphere, in Petri dishes, at 28 °C in the dark for six days. The pathogen infection thread (a dark filament) within the root is marked in the photos by a black line near each root. Control—root inoculated with *M. maydis* without fungicides. (**C**) Effect of Azoxystrobin seed coating (0.002 cm^3^/seed) on plant development in a greenhouse (adapted from [28]). A photograph of all the plants’ aboveground parts (upper panel) and leaves (lower panel) in the treatment and the non-protected control groups 72 days after sowing (DAS), 13 days after fertilization (DAF). (**D**) Late wilt disease symptoms in a field experiment 71 DAS, 16 DAF (adapted from [27]). Upper panel—wilt symptoms of a non-protected plot (control) compared to an Azoxystrobin-treated plot. Lower panel—representative plants. The fungicide was applied in seed coating (0.002 cm^3^/seed, Azoxystrobin + Difenoconazole, “Ortiva-Top”, Syngenta, Basel, Switzerland, supplier Adama Makhteshim, Airport City, Israel) and at three intervals, 18, 31, and 45 DAS, 2.25 L/hectare. Disease symptoms include drying-out that progresses upwards in the plant, stem and leaf yellowing, and dehydration.

**Figure 7 pathogens-11-00013-f007:**
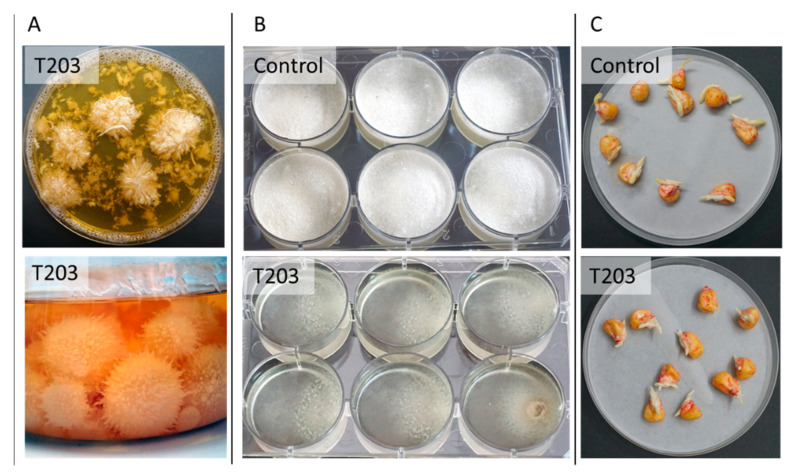
In vitro estimation of *Trichoderma asperelloides* (T203)-secreted metabolites-based biological control against *Magnaporthiopsis maydis* (adapted from [108]). (**A**) T203-submerged cultures grown with shaking (150 rpm) for isolating secreted metabolites. (**B**) Static shallow media cultures of *M. maydis* on potato dextrose broth (PDB) medium containing T203-secreted metabolites filtrate. Control is PDB medium *M. maydis* cultures maintained under the same conditions. (**C**) Effect of growth media of T203 isolate on corn seed germination. The seeds were germinated in Petri dishes soaked in 4 mL of PDB (control) or PDB + secretion products (growth medium filtrate six days after T203 growth). All images are displayed after 5–6 days incubation at 28 ± 1 °C in the dark.

**Figure 8 pathogens-11-00013-f008:**
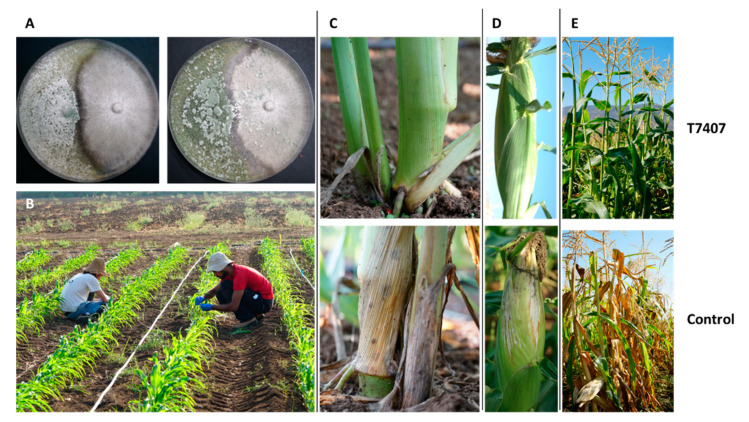
*Trichoderma longibrachiatum* (T7407) biological control against *Magnaporthiopsis maydis* in the lab and the field (adapted from [58,108]). (**A**) Plate mycoparasitism assay to identify interactions between *Magnaporthiopsis maydis* and T7407 in a potato dextrose agar (PDA)-rich medium. The two fungi were placed opposite each other, T7407 on the left and *M. maydis* on the right. Photos were taken after 3 and 10 days of growth. (**B**) Field inoculation of 20-day-old seedlings by an *M. maydis*-infected toothpick. The toothpicks were used for stabbing each plant at the near-surface portion of the stem. (**C**) The lower stem (first aboveground internode) disease symptoms. (**D**) The cobs’ spathes disease symptoms. (**E**) The experiment’s plots. Representative images of the field plants were taken 82 days after sowing. Controls are unprotected diseased plants.

## Data Availability

The data presented in this study are available on request from the corresponding author.

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
