# Peer review of "Control Strategies to Cope with Late Wilt of Maize"

_pathogens, 2021, doi:10.3390/pathogens11010013_

Round 1
Reviewer 1 Report
Control Strategies to Cope with Late Wilt of Maize is a 26-page review article. The manuscript is clear, comprehensive, and presented in a well-structured manner.
As the significance of the pathogen increases, this publication may be of great importance to scientists looking for information about the Late Wilt of Maize.
All the updated information regarding the pathogen M. maydis can be found in the manuscript, including recent information on the fungus lifecycle, primary and secondary hosts, interactions with other phytopathogens and biocontrol agents, and the fungus’ ability to survive under different environmental conditions.
Below I present my comments, that do not have any significant impact on the article, which in my opinion is exceptionally well written.
I believe it would be good to cite http://www.indexfungorum.org/names/Names.asp when mentioning the current name of the pathogen
Chapters 3.3.1 and 3.3.2 – please use cursive for Latin names of the species you are mentioning
In ‘References’ some more information is needed in 7,12 and 16.
Author Response
Responses to Reviewer 1’s comments
I thank the reviewer for investing substantial efforts, which is contributing to this manuscript. The remarks and suggestions improved this scientific paper. Your contribution is greatly appreciated.
General comments:
Control Strategies to Cope with Late Wilt of Maize is a 26-page review article. The
manuscript is clear, comprehensive, and presented in a well-structured manner.
As the significance of the pathogen increases, this publication may be of great importance to scientists looking for information about the Late Wilt of Maize.
All the updated information regarding the pathogen M. maydis can be found in the manuscript, including recent information on the fungus lifecycle, primary and secondary hosts, interactions with other phytopathogens and biocontrol agents, and the fungus’ ability to survive under different environmental conditions.
Below I present my comments, that do not have any significant impact on the article, which in my opinion is exceptionally well written.
Thank you for the constructive and essential comments. All your comments and suggestions were accepted and corrected, as detailed in the following point-by-point response.
Specific comments
I believe it would be good to cite http://www.indexfungorum.org/names/Names.asp when mentioning the current name of the pathogen
Thank you for this suggestion. The information was added to the text (lines 92-97): “The taxonomic tree of this fungus is Phylum: Ascomycota, Subphylum: Pezizomycotina, Class: Sordariomycetes, Subclass: Sordariomycetidae, Family: Magnaporthaceae, Genus: Magnaporthiopsis, Species: Magnaporthiopsis maydis (the most updated scientific name of the pathogen [38,39], Index Fungorum database, website: http://www.indexfungorum.org/names/NamesRecord.asp?RecordID=810225).”
Chapters 3.3.1 and 3.3.2 – please use cursive for Latin names of the species you are mentioning
All species’ Latin names were corrected to italic, as advised by the reviewer.
In ‘References’ some more information is needed in 7,12 and 76.
The missing information was added.
Reviewer 2 Report
Good job. The paper can certainly be a starting point for future research, especially where the pathogen has not yet been reported. Still, indeed, in the face of climate change, its incidence is imminent.
Author Response
Responses to Reviewer 2’s comments
Good job. The paper can certainly be a starting point for future research, especially where the pathogen has not yet been reported. Still, indeed, in the face of climate change, its incidence is imminent.
I want to express my sincere appreciation to the reviewer. The time and effort invested are greatly appreciated. Thank you.
Reviewer 3 Report
The author presents a review dealing wih control strategies helping to combat Late Wilt of maize. He recently reviewed general aspects of the disease in another review (Degani et al., 2021). Both reviews are clearly referred to each other, both are clearly distinguished and present seperate data and information, there are no big overlaps. Therefore the second review on this topic with another focus makes sense, since all aspects being covered in one review would have been too much of information.
The author is an expert in this disease and presents a well-written, comprehensive ad clearly structured review on Late Wilt of maize, including some of his own recent research results.
370, 392, 476, 482 and other positions until line 527: organism names are not in italics, please correct this.
Author Response
Responses to Reviewer 3’s comments
I want to express my sincere appreciation to the reviewer. The time and effort invested are greatly appreciated. Thank you.
General comments:
The author presents a review dealing wih control strategies helping to combat Late Wilt of maize. He recently reviewed general aspects of the disease in another review (Degani et al., 2021). Both reviews are clearly referred to each other, both are clearly distinguished and present seperate data and information, there are no big overlaps. Therefore the second review on this topic with another focus makes sense, since all aspects being covered in one review would have been too much of information.
The author is an expert in this disease and presents a well-written, comprehensive, and clearly structured review on Late Wilt of maize, including some of his own recent research results.
Thank you for the constructive and essential comments. Your comments and suggestions were accepted and corrected, as detailed below.
Specific comments
370, 392, 476, 482, and other positions until line 527: organism names are not in italics. Please correct this.
All species’ Latin names were corrected to italic, as advised by the reviewer.